## Clinical characteristics and outcomes of hospitalized patients with COVID-19 in a city of South Brazil: Have they changed through the first year of the pandemic?

**Henrique Demeneck***, **André Luiz Parmegiani de Oliveira, Julia do Carmo Machado Kneipp Lopes, Leonardo Ryoiti Matsunago, Luiza Cavalca Grupenmacher, Marcos Roberto Curcio Pereira, Rebecca Benício Stocco, Salma Ali El Chab Parolin, Marcia Olandoski, Cristina Pellegrino Baena**

Postgraduate Program in Health Sciences, Pontifícia Universidade Católica do Paraná, Curitiba, Brazil

* drhenriquedemeneck@gmail.com

## Abstract

The Coronavirus Disease 2019 (COVID-19) pandemic posed various challenges to the healthcare system and disease management. This study aimed to describe changes in the clinical characteristics and outcomes of hospitalized patients during the first year of the COVID-19 pandemic in a city in southern Brazil. This prospective study was carried out in two tertiary care private hospitals in Curitiba. A total of 1151 patients hospitalized between March 2020 and March 2021 were included. We identified three epidemiological critical periods of the pandemic and compared patients' characteristics and the frequencies of oral intubation, intensive care unit (ICU) admission and mortality. Continuous variables were analyzed by variance analysis model (ANOVA) or the Kruskal–Wallis nonparametric test and categorical variables by the chi-square or Fisher's exact test. Models for univariate and multivariate logistic regression analysis were adjusted to identify the factors associated with mortality. All p-values were two-tailed and $p < 0.05$ was considered statistically significant. The average age of the patients was 58 years and 60.9% (n = 701) were males. The most prevalent comorbidities were systemic arterial hypertension, diabetes and obesity. There were no significant variations in the demographic characteristics and previous comorbidities of the patients for the different periods of analysis. Mortality was positively associated with the age ≥65 years and the presence of one or more cardiometabolic comorbidities ($p < 0.001$). March 2021 was the most important critical period of the pandemic since there were higher frequencies of patients admitted later in the course of the disease, with desaturation and more symptoms at hospital admission ($p < 0.001$). There was also an increase in the duration of hospital stay ($p < 0.001$) and the frequencies of all critical outcomes for this period: oral intubation ($p < 0.001$), ICU admission (p = 0.606) and mortality (p = 0.001). Our key findings revealed that, although there were no statistically significant differences between the subgroups of hospitalized patients over time in terms of demographic characteristics and comorbidities, the course of COVID-19 was significantly more severe for patients admitted to the hospital at the end of the first year of the pandemic in Brazil.

**Data Availability Statement:** All relevant data are within the paper and its Supporting Information files.

**Funding:** The author(s) received no specific funding for this work.

**Competing interests:** The authors have declared that no competing interests exist.

# Introduction

The Coronavirus Disease 2019 (COVID-19) was declared a pandemic by the World Health Organization in March 2020 [1]. For most patients, the infection was mild to moderate; however, about 15% of them were hospitalized due to complications such as respiratory impairment and multiple organ dysfunctions [2, 3].

In the initial series of 138 hospitalized patients described in January 2020 in Wuhan, China, the overall mortality was 4.3% and severe cases were more prevalent among older males with comorbidities [4]. Subsequent studies identified the duration of the symptoms, the occurrence of desaturation and/or dyspnea and the need for supplementary oxygen as important prognostic factors [5, 6].

Studies conducted in other countries confirmed that patients aged ≥65 years and those with comorbidities (obesity, systemic arterial hypertension and diabetes) have an increased risk of intensive care unit (ICU) admission, prolonged duration of hospital stay and mortality [7–9].

The rates of hospital admission due to the disease have varied across the country and through the time during the COVID-19 pandemic. Up to January 2023, Brazil registered more than 36 million cases of COVID-19 and 695.000 deaths [10]. Although previous studies have identified the clinical characteristics of hospitalized patients in the northeast and southeast Brazil, it has not been elucidated whether these characteristics varied through the first year of the pandemic in the country and if the time of occurrence of the hospitalization represented an independent risk factor for negative outcomes [11, 12].

We aim to answer these questions as these variations may reflect changes in the prognostic factors, healthcare system burden and host-pathogen interactions at different periods of the pandemic in Brazil. Tracking these features may lead to a better understanding of how the clinical course of COVID-19 varied in our community and which temporal factors may have contributed to variations in the epidemiology of the disease at the local level.

# Methods

This is a prospective study carried out in two tertiary care private hospitals located in Curitiba, South Brazil: Marcelino Champagnat Hospital and Nossa Senhora das Graças Hospital. Patients hospitalized between March 2020 and March 2021 and who met the following study inclusion criteria were enrolled: age >18 years; COVID-19 infection confirmed by clinical-radiological presentation plus a nasopharyngeal swab polymerase chain reaction (PCR) positive to SARS-CoV2 realized within 14 days upon hospital admission (Cobas® SARS-CoV-2 Test, Roche Molecular Systems, Branchburg, NJ, United States); and voluntary participation confirmed by the signing of written informed consent by the patient or by a legal guardian of severely ill patients. An epidemiological review of the medical registries was carried out using a formulary designed to obtain information regarding the demographic, clinical and evolution of the cases (age, sex, comorbidities, continuous medication, duration of hospital stay, need for mechanical ventilation, frequency of ICU admissions, the occurrence of complications and death).

Each one of the three main periods of analysis (labeled as critical periods 1, 2 and 3) was determined from the identification of the date on which the highest absolute number of deaths due to COVID-19 was recorded in Curitiba according to local epidemiological bulletins. Each of these critical periods was defined as a period of 28 consecutive days including the two weeks before and the two weeks after the identified date. All the other periods were denominated as non-critical (Fig 1).

For the analysis, cardiometabolic comorbidity was defined as the presence of at least one of the following conditions: systemic arterial hypertension, diabetes mellitus, obesity, chronic cardiac insufficiency, chronic coronary syndrome, dyslipidemia, previous acute myocardial infarction, and previous cerebrovascular accident. The surgeries considered in data collection were cardiac catheterization, chest drainage, tracheostomy and/or limb amputation.

Data were compiled using SPSS v.28.0 (IBM, Armonk, NY) and analyzed by GraphPad Prism v7 for Windows (GraphPad, San Diego, CA). Continuous variables were expressed as median values and analyzed by analysis of variance (ANOVA) or the Kruskal–Wallis nonparametric test. Categorical variables were expressed as absolute frequencies and proportions and analyzed by chi-square or Fisher's exact test. Models for univariate and multivariate logistic regression analysis were adjusted to identify the factors associated with mortality. For the multivariate model, any variable with $p < 0.05$ in the univariate analyses was included. The Wald test was used to assess the significance of each variable included in the models and the measure of association used was the odds ratio (OR) with confidence intervals (CI) of 95%. All p-values were two-tailed, and a value of $p < 0.05$ was considered statistically significant. This study was approved by the Research Ethics Committee of the institutions to which it is affiliated (number: 30188020.7.1001.0020).

## Results

According to the local official bulletins, during the first year of the pandemic in Curitiba, the dates of occurrence of the highest absolute number of deaths by COVID-19 in the city were Aug 5th 2020 (27 deaths, 733 new confirmed cases), Dec 13th 2020 (33 deaths, 2.269 new confirmed cases) and March 15th 2021 (57 deaths, 2.757 new confirmed cases). A total of 1151 consecutive patients who were hospitalized during the first year of the pandemic and met inclusion criteria were analyzed, totalling to following numbers for each period: 120 (critical period 1), 120 (critical period 2), 149 (critical period 3) and 762 patients (for all non-critical periods). A total of 42 patients with incomplete medical information and those who did not agree to participate in the study were excluded.

There were no significant differences between the subgroups of hospitalized patients with respect to sex, age and the frequencies of the comorbidities analyzed through time. However, the third critical period (March of 2021) showed a more severe course of COVID-19 with prolonged hospitalization, higher relative frequencies of ICU admissions, oral intubation, hospital interventions (such as fibro-bronchoscopy and surgeries) and mortality. In addition, during the third critical period, the patients were often admitted later on in the course of the disease (with more than 7 days of symptoms) and had a greater number of symptoms at hospital admission with higher frequencies of desaturation (pulse oximetry below 95%).

The average age of hospitalized patients was 58 [±17,1] years and 60.9% (n = 701) were males. The most prevalent comorbidities were systemic arterial hypertension (47.7%, n = 549), diabetes (24.3%, n = 280) and obesity (39.8%, n = 392). Their general clinical characteristics are summarized in Table 1.

ICU admission, oral intubation and mortality occurred in 42.9% (n = 493), 27.7% (n = 317) and 17.7% (n = 204) of the hospitalized patients, respectively. The highest frequencies of oral intubation (40.5%) and mortality (29.5%) occurred in critical period 3, and the rates of admission to the ICU remained high during the period. The median duration of hospitalization was 8 [0–214] days and there was a statistically significant increase in the duration of hospitalizations in the critical period 3 with a median duration of 10 days for this period (p<0.001). The frequencies and variations through the time of these outcomes are shown in Table 2 and Fig 2.

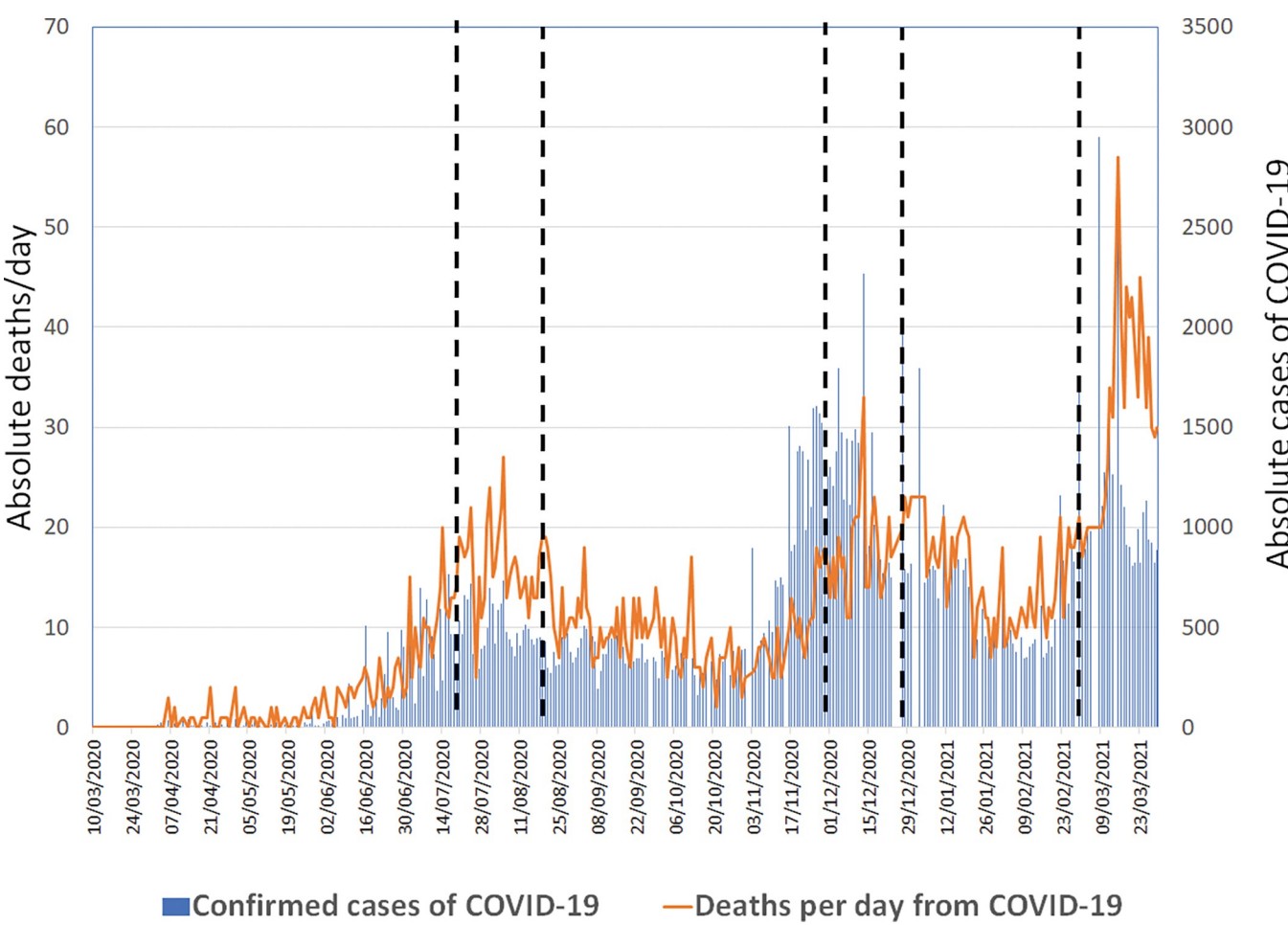

**Fig 1. Number of confirmed cases and daily obits due to COVID-19 in Curitiba through the first year of the pandemic according to official local epidemiological bulletins.** The three critical periods of occurrence of the cases are shown in the figure.

In the univariate analyses, mortality was positively associated with the age $\geq 65$ years (OR = 5.51; 95% CI = 3.94–7.71, p<0.001), the presence of at least one cardiometabolic comorbidity (OR = 4.57; 95% CI = 3.07–6.80, p<0.001) and the occurrence of critical hospital events, such as the need for surgery (OR = 3.49; 95% CI = 2.21–5.51, p<0.001), ICU admission (OR = 23.9; 95% CI = 13.9–41.1, p<0.001) and oral intubation (OR = 35.7; 95% CI = 23.1–55.3, p<0.001).

The multivariate analysis showed a positive and independent association of mortality with the age $\geq 65$ years (OR = 1.06; 95% CI = 1.05–1.08, p<0.001), the presence of two or more comorbidities (OR = 1.71; 95% CI = 1.01–2.90, p = 0.045), and admission at the critical period 3 when compared to the critical periods 1 (OR = 0.35; 95% CI = 0.16–0.76, p = 0.008) and 2 (OR = 0.48; 95% CI = 0.23–0.98, p = 0.044) of the pandemic (Fig 3).

## Discussion

The most frequent clinical characteristics of our population were the male sex and the presence of cardiometabolic comorbidities. Previously, Domingos Correa *et al.* analyzed the characteristics of 1296 patients admitted to ICU due to COVID-19 during the first year of the pandemic in Brazil in a private hospital in Sao Paulo and found similar features among hospitalized patients [13]. Moreover, Vahey *et al.* described that chronic hypoxemic respiratory failure with

**Table 1. General clinical characteristics of the patients hospitalized due to COVID-19.** Values are reported as absolutes numbers and relative frequencies.

| Variables | n (1151) | % |
|---|---|---|
| Age (years) ≥65 | 455 | 39.50% |
| Males | 701 | 60.90% |
| Systemic arterial hypertension | 549 | 47.70% |
| Diabetes Mellitus | 280 | 24.30% |
| Body Mass Index (BMI) ≥ 30 | 392 | 39.80% |
| Chronic heart failure | 55 | 4.80% |
| Chronic coronary syndrome | 81 | 7.00% |
| Previous acute myocardial infarction | 47 | 4.10% |
| Previous cerebrovascular accident | 31 | 2.70% |
| Chronic peripheral arterial syndrome | 20 | 1.70% |
| Arrhythmia | 63 | 5.50% |
| Asthma | 47 | 4.10% |
| Chronic obstructive pulmonary disease (COPD) | 47 | 4.10% |
| Chronic kidney disease | 53 | 4.60% |
| Autoimmune disease | 35 | 3.00% |
| Cancer | 63 | 5.50% |
| Dyslipidemia | 268 | 23.30% |
| Cardiometabolic disease* | 684 | 59.40% |
| Number of comorbidities: | | |
| 0 or 1 | 586 | 50.90% |
| 2 or more | 565 | 49.10% |
| Number of symptoms at hospital admission ≥4 | 506 | 44.00% |
| Days with symptoms before hospital admission > 7 | 567 | 51.50% |
| Oxygen saturation at hospital admission < 95% | 705 | 62.90% |

*At least one out of: systemic hypertension, *diabetes mellitus*, obesity, cardiac insufficiency, chronic coronary syndrome, dyslipidemia, previous acute myocardial infarction, and previous cerebrovascular accident.

oxygen requirement, obesity, hypertension and male sex were factors associated with hospitalization due to COVID-19 [14]. The clinical characteristics of our patients are compatible with those shown by previous studies with similar methodologies conducted in other countries and Brazil [7, 11].

There was no significant variation in the age and the previous comorbidities of the hospitalized patients for the different periods of the analysis. The average age of 58 years was compatible with a previous study conducted by Lorenz *et al.* in Southeast Brazil that analyzed more than 300 thousand severe cases of hospitalized patients with COVID-19 in the state of Sao Paulo and found a similar average age [15]. Although the average age of our sample was lower than 60 years, the prevalence of patients aged ≥65 years was higher among patients admitted to ICU, which is also in accordance with previous studies [11].

For our population, mortality was more frequent among older patients with comorbidities and for patients who needed ICU admission and/or oral intubation. In addition, the risk of mortality was higher during the third critical period. Previous studies also identified that March 2021 was as a critical period of the pandemic in Brazil as it was associated with a collapse of the healthcare system [11, 12].

Previous studies conducted in cities located in northeast and southeast Brazil showed an increased incidence and lethality rates due to COVID-19 in the first trimester of 2021.

**Table 2. Clinical characteristics and outcomes for each period of the analysis—patients are grouped according to the period of hospital admission.**

| Variables | Period | | | | | | p** |
|---|---|---|---|---|---|---|---|
| | Non critical (N = 263) | Critical period 1 (N = 120) | Non critical (N = 286) | Critical period 2 (N = 120) | Non critical (N = 213) | Critical period 3 (N = 149) | |
| **Clinical characteristics** | | | | | | | |
| Age (years) ≥65 | 104 (39.5%) | 42 (35%) | 120 (42%) | 47 (39.2%) | 85 (39.9%) | 57 (38.3%) | 0.869 |
| Males | 152 (57.8%) | 66 (55%) | 181 (63.3%) | 76 (63.3%) | 135 (63.4%) | 91 (61.1%) | 0.5 |
| Arterial systemic hypertension | 108 (41.1%) | 56 (46.7%) | 143 (50%) | 57 (47.9%) | 108 (50.7%) | 77 (51.7%) | 0.219 |
| Diabetes mellitus | 77 (29.3%) | 23 (19.2%) | 56 (19.6%) | 30 (25.2%) | 60 (28.2%) | 34 (22.8%) | 0.058 |
| Body Mass Index (BMI) ≥ 30 | 69 (34%) | 43 (40.6%) | 104 (40.2%) | 54 (49.1%) | 72 (37.7%) | 50 (43.1%) | 0.171 |
| Number of symptoms at hospital admission ≥4 | 111 (42.2%) | 57 (47.5%) | 112 (39.3%) | 41 (34.2%) | 118 (55.7%) | 67 (45%) | **0.001** |
| Days with symptoms before hospital admission >7 | 106 (40.5%) | 63 (52.5%) | 140 (49.1%) | 72 (60%) | 110 (55.6%) | 76 (65%) | **<0.001** |
| Oxygen saturation at hospital admission < 95% | 132 (54.8%) | 54 (45%) | 168 (59.2%) | 87 (74.4%) | 154 (72.3%) | 110 (75.3%) | **<0.001** |
| **In-hospital outcomes** | | | | | | | |
| Fibro-bronchoscopy | 5 (1.9%) | 3 (2.5%) | 18 (6.3%) | 7 (5.8%) | 20 (9.4%) | 25 (16.8%) | **<0.001** |
| Chest drainage | 6 (2.3%) | 0 (0%) | 5 (1.8%) | 7 (5.8%) | 9 (4.2%) | 10 (6.7%) | **0.007** |
| Surgery during hospital stay* | 12 (4.6%) | 5 (4.2%) | 22 (7.7%) | 16 (13.3%) | 15 (7%) | 18 (12.1%) | **0.009** |
| Vasoactive drug | 54 (20.5%) | 20 (16.8%) | 64 (22.4%) | 29 (24.2%) | 54 (25.4%) | 52 (34.9%) | **0.008** |
| Deep vein thrombosis (DVT) | 0 (0%) | 0 (0%) | 4 (1.4%) | 1 (0.8%) | 1 (0.5%) | 5 (3.4%) | **0.016** |
| Pulmonary embolism | 6 (2.3%) | 4 (3.4%) | 8 (2.8%) | 2 (1.7%) | 12 (5.6%) | 15 (10.1%) | **0.001** |
| Oral intubation | 69 (26.2%) | 20 (16.8%) | 65 (22.9%) | 36 (30.3%) | 67 (31.5%) | 60 (40.5%) | **<0.001** |
| Intensive care unit (ICU) admission | 107 (40.7%) | 46 (38.3%) | 123 (43%) | 57 (47.5%) | 90 (42.5%) | 70 (47%) | 0.606 |
| Death | 44 (16.7%) | 16 (13.3%) | 39 (13.6%) | 20 (16.7%) | 41 (19.3%) | 44 (29.5%) | **0.001** |

*cardiac catheterization, chest drainage, tracheostomy and/or limb amputation.

** comparisons between the six periods using chi-squared test.

However, these studies did not focus on the variations in the clinical characteristics and outcomes of the patients through the pandemic. A small study conducted by Nascimento *et al.* with 142 patients diagnosed with COVID-19 and admitted to ICU in South Brazil also suggested that patients admitted in 2021 showed more severe outcomes with higher Sequential Organ Failure Assessment Score (SOFA) and higher rates of ICU admissions, oral intubation, and mortality [16].

Of note, the median duration of hospital stay has increased through the course of the pandemic. The median duration of hospital stay increased to 10 days in critical period 3 in comparison to 5.5 and 7 days in critical periods 1 and 2, respectively. Other studies have suggested that the duration of hospital stay tends to be longer among patients admitted later on the course of the disease and with more severe presentation upon hospital admission [12]. It could be a possible explanation for the increased duration of hospital stay found for our sample.

Furthermore, the variant of concern (VOC) gamma or P.1 of SARS-CoV 2 has emerged around mid-November 2020 and was first described in Manaus, Brazil, in early January 2021. The emergence of gamma has probably influenced our findings, as it exhibited increased transmissibility and immune evasion capacity and was critical to accelerate the pandemic in Brazil [17, 18].

One limitation of our study is it included only private hospitals. A previous study conducted in Southeast Brazil showed that higher mortality rates were observed among patients

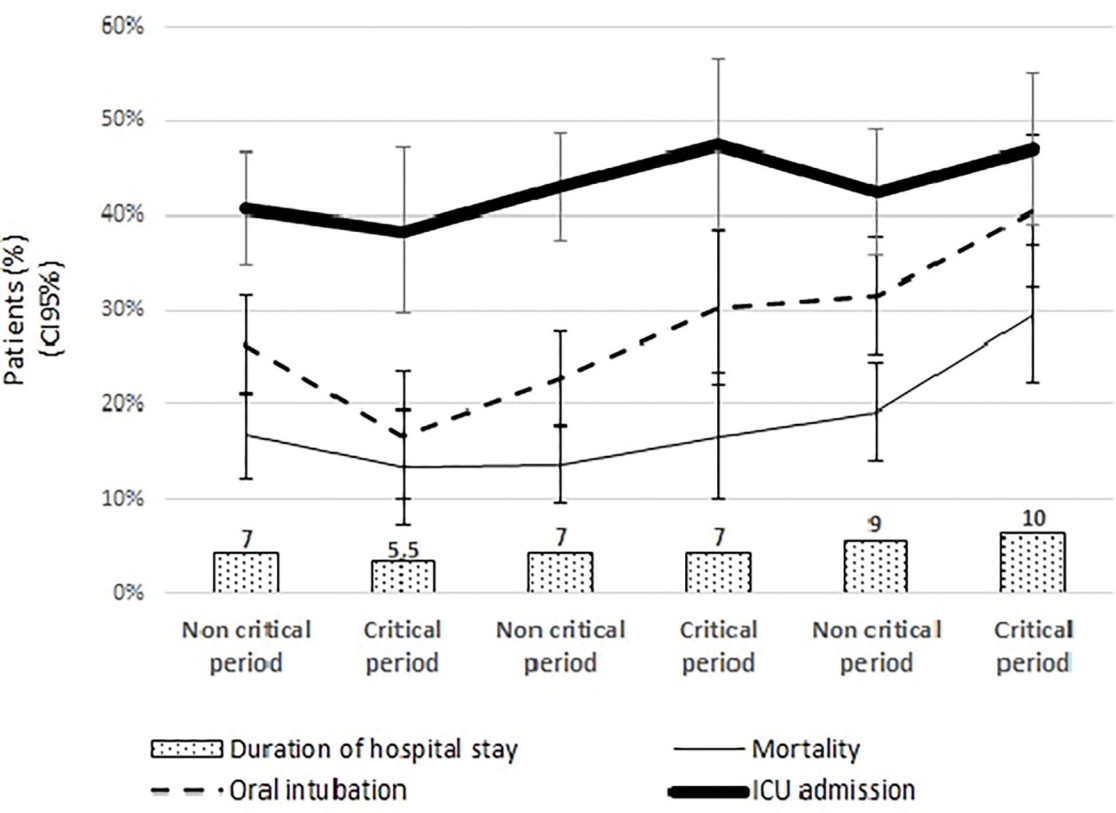

**Fig 2. Frequencies of critical outcomes between hospitalized patients through the first year of the pandemic.** Duration of hospital stay is expressed as the median of the absolute number of days whereas oral intubation, mortality and intensive care unit (ICU) admission are expressed as relative frequencies. CI, confidence interval.

with COVID-19 admitted to public hospitals [19]. In the analysis of the factors associated with the deaths of 420 individuals hospitalized with COVID-19 in the state of Espirito Santo, Brazil, Maciel *et al.* showed that the hospital mortality was higher for the cases notified by public institutions with an odds ratio of 8.23 with a 95% CI [20]. Thus, our data could have underestimated the real severity of the hospitalized patients in the community during each period, since only private institutions were included. Moreover, we have not related our findings to variations on the dominant virus' variants and the impacts of COVID-19 on the healthcare system over time. However, official genomic vigilance reported in our location shows that there was a predominance of lineage P.2 from December 2020 to January 2021 which was followed by the dominance of the VOC Gamma in February and March 2021. As the Gamma variant spreads faster and was more transmissible, such variations of the predominant lineage of SARS-CoV2 may have influenced our findings of the different critical periods analyzed [17].

One strength of this study is that it analysed the characteristics of hospitalized patients in the pre-vaccination phase of the pandemic. It can contribute to the elucidation of the impacts of mass vaccination campaigns for the control of COVID-19 transmission in the community. In addition, our study included a significant number of patients hospitalized due to COVID-19 in a city in southern Brazil and the number of publications with a similar methodology for this population is limited.

Our report identified the variations in the risk and prognostic factors of COVID-19 among hospitalized patients, providing meaningful clinical insights for a better understanding of the

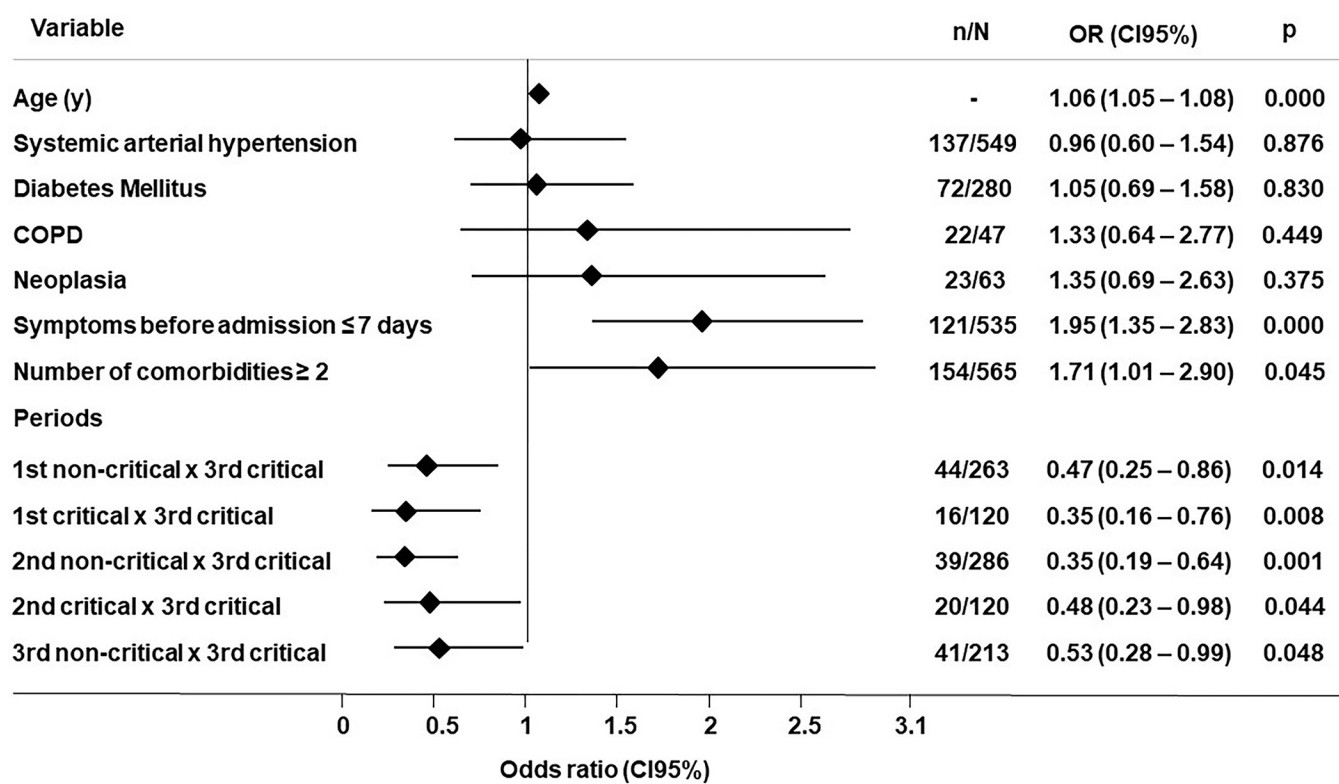

| Variable | | n/N | OR (CI95%) | p |
|---|---|---|---|---|
| Age (y) | | - | 1.06 (1.05 – 1.08) | 0.000 |
| Systemic arterial hypertension | | 137/549 | 0.96 (0.60 – 1.54) | 0.876 |
| Diabetes Mellitus | | 72/280 | 1.05 (0.69 – 1.58) | 0.830 |
| COPD | | 22/47 | 1.33 (0.64 – 2.77) | 0.449 |
| Neoplasia | | 23/63 | 1.35 (0.69 – 2.63) | 0.375 |
| Symptoms before admission ≤ 7 days | | 121/535 | 1.95 (1.35 – 2.83) | 0.000 |
| Number of comorbidities ≥ 2 | | 154/565 | 1.71 (1.01 – 2.90) | 0.045 |
| Periods | | | | |
| 1st non-critical x 3rd critical | | 44/263 | 0.47 (0.25 – 0.86) | 0.014 |
| 1st critical x 3rd critical | | 16/120 | 0.35 (0.16 – 0.76) | 0.008 |
| 2nd non-critical x 3rd critical | | 39/286 | 0.35 (0.19 – 0.64) | 0.001 |
| 2nd critical x 3rd critical | | 20/120 | 0.48 (0.23 – 0.98) | 0.044 |
| 3rd non-critical x 3rd critical | | 41/213 | 0.53 (0.28 – 0.99) | 0.048 |

**Fig 3. Multivariate logistic regression model of the factors associated with mortality—Wald test, p<0.05.** OR, odds ratio; CI, confidence interval; COPD, chronic obstructive pulmonary disease.

host-pathogen interactions during different periods of the pandemic. Even though healthcare workers gained knowledge about the management of COVID-19 during the first two critical moments of the pandemic, the prognosis and clinical presentation of the disease worsened in the third studied period. This could be attributed to an overload of challenges involving deterioration of the prognosis and to the emergence of mutated viral strains due to the absence of vaccination in the population. Thus, our findings show that it is important to consider the variable time of hospital admission as an independent prognostic factor for mortality due to COVID-19 among hospitalized patients.

Furthermore, our study may help local authorities to analyze the impact of different measures adopted over time such as restriction of services and the adoption of isolation measures, for controlling the transmission of the disease. Future work will aim to predict which variables (i.e. predominantly circulating virus strains, vaccination status of the population and healthcare system burden) may have influenced the variations in the epidemiology of COVID-19 in the local community and thus may have contributed to our findings. These learnings can contribute to the better management of a possible future pandemic.

We described variations of the clinical characteristics and outcomes of hospitalized patients during the first three critical moments of the incidence of the disease in a city in southern Brazil. Interestingly, although there were no significant differences in the previous clinical characteristics of the patients, severe clinical presentations at hospital admission and negative outcomes such as mortality were more frequent in March 2021. More studies are necessary to elucidate which factors showed a cause-effect relationship to explain such findings.

## Supporting information

**S1 File. Data related to the patients are included in a separate file uploaded as a "Supporting information file".**
(XLSX)

## Author Contributions

**Conceptualization:** Henrique Demeneck, André Luiz Parmegiani de Oliveira, Rebecca Benício Stocco, Salma Ali El Chab Parolin, Marcia Olandoski, Cristina Pellegrino Baena.

**Data curation:** Henrique Demeneck, André Luiz Parmegiani de Oliveira, Julia do Carmo Machado Kneipp Lopes, Leonardo Ryoiti Matsunago, Luiza Cavalca Grupenmacher, Marcos Roberto Curcio Pereira, Rebecca Benício Stocco, Salma Ali El Chab Parolin, Marcia Olandoski, Cristina Pellegrino Baena.

**Formal analysis:** Henrique Demeneck, Salma Ali El Chab Parolin, Marcia Olandoski, Cristina Pellegrino Baena.

**Investigation:** Henrique Demeneck, Salma Ali El Chab Parolin, Marcia Olandoski, Cristina Pellegrino Baena.

**Methodology:** Henrique Demeneck, André Luiz Parmegiani de Oliveira, Julia do Carmo Machado Kneipp Lopes, Leonardo Ryoiti Matsunago, Luiza Cavalca Grupenmacher, Marcos Roberto Curcio Pereira, Rebecca Benício Stocco, Salma Ali El Chab Parolin, Marcia Olandoski, Cristina Pellegrino Baena.

**Project administration:** Henrique Demeneck, Cristina Pellegrino Baena.

**Software:** Henrique Demeneck, Marcia Olandoski, Cristina Pellegrino Baena.

**Supervision:** Marcos Roberto Curcio Pereira, Rebecca Benício Stocco, Marcia Olandoski, Cristina Pellegrino Baena.

**Validation:** Cristina Pellegrino Baena.

**Writing – original draft:** Henrique Demeneck, Marcia Olandoski, Cristina Pellegrino Baena.

**Writing – review & editing:** Henrique Demeneck, Cristina Pellegrino Baena.

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
