## [Decision Letter · Decision Letter 0]

20 Mar 2023

PONE-D-23-02590Clinical characteristics and outcomes of hospitalized patients with COVID-19 in a city of south Brazil: have they changed through the first year of the pandemic?PLOS ONE

Dear Dr. DEMENECK,

Thank you for submitting your manuscript to PLOS ONE. After careful consideration, we feel that it has merit but does not fully meet PLOS ONE’s publication criteria as it currently stands. Therefore, we invite you to submit a revised version of the manuscript that addresses the points raised during the review process.

We look forward to receiving your revised manuscript.

Kind regards,

Tariq Jamal Siddiqi

Academic Editor

PLOS ONE

Reviewers' comments:

Reviewer's Responses to Questions

**Comments to the Author**

1. Is the manuscript technically sound, and do the data support the conclusions?

Reviewer #1: Yes

2. Has the statistical analysis been performed appropriately and rigorously? 

Reviewer #1: Yes

3. Have the authors made all data underlying the findings in their manuscript fully available?

Reviewer #1: Yes

4. Is the manuscript presented in an intelligible fashion and written in standard English?

Reviewer #1: Yes

5. Review Comments to the Author

Reviewer #1: Demeneck et al. conducted a prospective study titled: “Clinical characteristics and outcomes of hospitalized patients with COVID-19 in a city of south Brazil: have they changed through the first year of the pandemic?”, in which the authors described the potential changes of the clinical characteristics and outcomes of hospitalized patients during the first year of the COVID-19 pandemic in a city of south Brazil. In my opinion the manuscript can be improved by inculcating the following changes:

1. Abstract: although the authors have summarized the background, the aims, the methods, and results of their study in the abstract of the manuscript, it would improve the manuscript if the authors add a few lines regarding the conclusion of their study in the abstract as well.

2. Introduction section: although the authors have mentioned that few studies have focused on identifying the variation in clinical characteristics of hospitalized patients, it would improve the manuscript if the authors cite those studies and mention what is lacking in those studies and how this study hopes to fulfill that gap in knowledge. Furthermore, the authors should also consider adding a few lines regarding why the knowledge pertaining to variation in clinical characteristics is necessary in the first place.

3. Introduction section: “The present study investigated those characteristics and the frequencies of negative outcomes - oral intubation, ICU admission and mortality – among patients admitted to the hospital due to COVID-19 during the first year of the pandemic at two referral centers located in the city of Curitiba, south Brazil.”; the authors should consider removing these lines from the introduction section, as they are redundant.

4. Methods section: where the authors mention the names of the two hospitals, in which the study was conducted, the authors should also mention the name of the city in which the two hospitals are located.

5. Methods section: the number of included and excluded patients should be removed from the methods section of the manuscript and should be mentioned in the beginning of the results section.

6. Results section: in the methods section, the authors mentioned that for the multivariate logistic regression, sex, in addition to age and any variable with p < 0.10 in the univariate analyses were included. However, the results for the multivariate logistic regression with regards to sex have neither been reported in the manuscript nor in the corresponding figure.

7. Results section: the corresponding Odds Ratios with their respective 95% confidence intervals for each predictor variable should also be reported for the multivariate logistic regression analysis.

8. Discussion section: it would add value to the significance and relevance of the study if the authors expand upon the clinical implications of their findings in the discussion section of the manuscript.

9. The authors should consider thoroughly proofreading the manuscript, as typos and grammatical errors are present at certain places in the manuscript.

6. PLOS authors have the option to publish the peer review history of their article (what does this mean?). If published, this will include your full peer review and any attached files.

Reviewer #1: No

---

## [Author Response · Author response to Decision Letter 0]

12 Apr 2023

Thanks for the encouraging comments and the contributions to enhance the quality of our manuscript. We conducted a target review of the article and we have now clarified the points that needed attention according to your review. 

1. Abstract: although the authors have summarized the background, the aims, the methods, and results of their study in the abstract of the manuscript, it would improve the manuscript if the authors add a few lines regarding the conclusion of their study in the abstract as well.

Reply – Thanks for pointing out this issue. We agree that there is a lack of a conclusion for the abstract presented. We have now added it to our abstract section that reads: 

“Our key findings revealed that, although there were no statistically significant differences between the subgroups of hospitalized patients over time in terms of demographic characteristics and comorbidities, the course of COVID-19 was significantly more severe for patients admitted to the hospital at the end of the first year of the pandemic in Brazil”.

2. Introduction section: although the authors have mentioned that few studies have focused on identifying the variation in clinical characteristics of hospitalized patients, it would improve the manuscript if the authors cite those studies and mention what is lacking in those studies and how this study hopes to fulfill that gap in knowledge. Furthermore, the authors should also consider adding a few lines regarding why the knowledge pertaining to variation in clinical characteristics is necessary in the first place.

Reply – We have now cited previous studies conducted in other locations of Brazil that evaluated the clinical characteristics of affected patients during the first year of the pandemic. We have also pointed out the relevance and innovative features of our study and its clinical implications for a better understanding of the local epidemiology of COVID-19. 

“Although previous studies have identified the clinical characteristics of hospitalized patients in the northeast and southeast Brazil, it has not been elucidated whether these characteristics varied through the first year of the pandemic in the country and if the time of occurrence of the hospitalization represented an independent risk factor for negative outcomes. [11,12]

We aim to answer these questions as these variations may reflect changes in the prognostic factors, healthcare system burden and host-pathogen interactions at different periods of the pandemic in Brazil. Tracking these features may lead to a better understanding of how the clinical course of COVID-19 varied in our community and which temporal factors may have contributed to variations in the epidemiology of the disease at the local level”. 

3. Introduction section: “The present study investigated those characteristics and the frequencies of negative outcomes - oral intubation, ICU admission and mortality – among patients admitted to the hospital due to COVID-19 during the first year of the pandemic at two referral centers located in the city of Curitiba, south Brazil.”; the authors should consider removing these lines from the introduction section, as they are redundant. 

Reply – We have excluded the sentence from the introduction as suggested by the reviewer. 

4. Methods section: where the authors mention the names of the two hospitals, in which the study was conducted, the authors should also mention the name of the city in which the two hospitals are located.

Reply- We have added the location of the hospitals. 

“This is a prospective study carried out in two tertiary care private hospitals located in Curitiba, South Brazil: Marcelino Champagnat Hospital and Nossa Senhora das Graças Hospital”

5. Methods section: the number of included and excluded patients should be removed from the methods section of the manuscript and should be mentioned in the beginning of the results section.

Reply- This information has been added in the first paragraph of the results section of the manuscript:

“A total of 1151 consecutive patients who were hospitalized during the first year of the pandemic and met inclusion criteria were analyzed, totalling to following numbers for each period: 120 (critical period 1), 120 (critical period 2), 149 (critical period 3) and 762 patients (for all non-critical periods). A total of 42 patients with incomplete medical information and those who did not agree to participate in the study were excluded.”

6. Results section: in the methods section, the authors mentioned that for the multivariate logistic regression, sex, in addition to age and any variable with p < 0.10 in the univariate analyses were included. However, the results for the multivariate logistic regression with regards to sex have neither been reported in the manuscript nor in the corresponding figure.

Reply – Thank you for your diligent review. Indeed, as pointed out by the reviewer, sex was not included in the multivariate analysis since it did not have statistical significance in the univariate analysis. The previous sentence has now been replaced in the methods section: 

 “For the multivariate model, any variable with p < 0.05 in the univariate analyses was included”.

7. Results section: the corresponding Odds Ratios with their respective 95% confidence intervals for each predictor variable should also be reported for the multivariate logistic regression analysis.

Reply-The corresponding Odds Ratio and the 95% CI for each variable were added to the text. The corresponding values are also shown in the figure number 3 of the article. 

“The multivariate analysis showed a positive and independent association of mortality with the age ≥65 years (OR = 1.06; 95% CI = 1.05 – 1.08, p<0.001), the presence of two or more comorbidities (OR = 1.71; 95% CI = 1.01 – 2.90, p = 0.045), and admission at the critical period 3 when compared to the critical periods 1 (OR = 0.35; 95% CI = 0.16 – 0.76, p = 0.008) and 2 (OR = 0.48; 95% CI = 0.23 – 0.98, p= 0.044) of the pandemic (Figure 3).”

8. Discussion section: it would add value to the significance and relevance of the study if the authors expand upon the clinical implications of their findings in the discussion section of the manuscript.

 Reply - We now have discussed that the findings of our study identified the time of hospital admission as an important and independent prognostic factor for mortality due to COVID-19 among hospitalized patients. We cite the clinical relevance of such findings and suggest the role of future studies in improving our knowledge related to the epidemiology of the disease. Moreover, we have added that the knowledge pertaining to variation in clinical characteristics of the affected individuals is useful since new pandemics can occur in the future and even though healthcare workers have gained knowledge about the management of COVID-19 during the first two critical moments, the prognosis and clinical presentation of the disease deteriorated in the third studied period. 

 “Our report identified the variations in the risk and prognostic factors of COVID-19 among hospitalized patients, providing meaningful clinical insights for a better understanding of the host-pathogen interactions during different periods of the pandemic. Even though healthcare workers gained knowledge about the management of COVID-19 during the first two critical moments of the pandemic, the prognosis and clinical presentation of the disease worsened in the third studied period. This could be attributed to an overload of challenges involving deterioration of the prognosis and clinical presentation of the health system and to the emergence of mutated viral strains due to the absence of vaccination in the population. Thus, our findings show that it is important to consider the variable time of hospital admission as an independent prognostic factor for mortality due to COVID-19 among hospitalized patients. 

 Furthermore, our study may help local authorities to analyze the impact of different measures adopted over time such as restriction of services and the adoption of isolation measures, for controlling the transmission of the disease. Future work will aim to predict which variables (i.e. predominantly circulating virus strains, vaccination status of the population and healthcare system burden) may have influenced the variations in the epidemiology of COVID-19 in the local community and thus may have contributed to our findings. These learnings can contribute to the better management of a possible future pandemic.” 

 9. The authors should consider thoroughly proofreading the manuscript, as typos and grammatical errors are present at certain places in the manuscript.

 Reply- The manuscript was fully reviewed and proofread by Editage.

---

## [Decision Letter · Decision Letter 1]

19 May 2023

Clinical characteristics and outcomes of hospitalized patients with COVID-19 in a city of south Brazil: have they changed through the first year of the pandemic?

PONE-D-23-02590R1

Dear Dr. DEMENECK,

We’re pleased to inform you that your manuscript has been judged scientifically suitable for publication and will be formally accepted for publication once it meets all outstanding technical requirements.

Kind regards,

Tariq Jamal Siddiqi

Academic Editor

PLOS ONE

Additional Editor Comments (optional):

Reviewers' comments:

Reviewer's Responses to Questions

**Comments to the Author**

1. If the authors have adequately addressed your comments raised in a previous round of review and you feel that this manuscript is now acceptable for publication, you may indicate that here to bypass the “Comments to the Author” section, enter your conflict of interest statement in the “Confidential to Editor” section, and submit your "Accept" recommendation.

Reviewer #1: (No Response)

2. Is the manuscript technically sound, and do the data support the conclusions?

Reviewer #1: Yes

3. Has the statistical analysis been performed appropriately and rigorously? 

Reviewer #1: Yes

4. Have the authors made all data underlying the findings in their manuscript fully available?

Reviewer #1: Yes

5. Is the manuscript presented in an intelligible fashion and written in standard English?

Reviewer #1: Yes

6. Review Comments to the Author

Reviewer #1: (No Response)

7. PLOS authors have the option to publish the peer review history of their article (what does this mean?). If published, this will include your full peer review and any attached files.

Reviewer #1: No

---

## [Editor Report · Acceptance letter]

22 May 2023

PONE-D-23-02590R1 

Clinical characteristics and outcomes of hospitalized patients with COVID-19 in a city of south Brazil: have they changed through the first year of the pandemic? 

Dear Dr. Demeneck:

I'm pleased to inform you that your manuscript has been deemed suitable for publication in PLOS ONE. Congratulations! Your manuscript is now with our production department. 

Kind regards, 

on behalf of

Dr. Tariq Jamal Siddiqi 

Academic Editor

PLOS ONE